# Severe Recurrent COVID-Associated Pulmonary Aspergillosis: A Challenging Case

**DOI:** 10.3390/healthcare10122483

**Published:** 2022-12-08

**Authors:** Luigi Vetrugno, Gian Marco Anzellotti, Regina Frontera, Zoe Parinisi, Barbara Sessa, Cristian Deana, Salvatore Maurizio Maggiore

**Affiliations:** 1Department of Anesthesia, “SS Annunziata” Hospital, Via dei Vestini, 66100 Chieti, Italy; 2Department of Medical, Oral and Biotechnological Sciences, University of Chieti-Pescara, 66100 Chieti, Italy; 3Department of Anesthesia and Intensive Care, Health Integrated Agency of Friuli Centrale, Piazzale S.M. Della Misericordia, 15, 33100 Udine, Italy; 4Department of Innovative Technologies in Medicine and Dentistry, Gabriele d’Annunzio University of Chieti-Pescara, 66100 Chieti, Italy

**Keywords:** pulmonary aspergillosis, COVID-19, critically ill patient, mechanical ventilation, tracheostomy

## Abstract

We report a rare case of severe COVID-19-associated pulmonary aspergillosis presenting as invasive pulmonary aspergillosis and subsequently invasive tracheobronchial aspergillosis during hospitalization in a critically ill patient who developed a further Aspergillus infection after home discharge. He needed readmission to the ICU and mechanical ventilation. We therefore strongly encourage a high degree of attention to fungal complications, even after viral recovery and ICU discharge.

## 1. Introduction

Aspergillus spp. mainly affects patients with a compromised immune system [1]. Different clinical manifestations of Aspergillus spp. infection are possible. Invasive pulmonary aspergillosis (IPA) is the most severe form of Aspergillus spp. infection that occurs after fungus invasion into the bloodstream. Invasive tracheobronchial aspergillosis (IATB) is a rare phenotype of IPA characterized by the infection of tracheobronchial spaces [2]. IPA and IATB have been described as deadly complications of severe viral infections as well, including influenza and coronavirus disease (COVID-19) [3]. COVID-19-associated pulmonary aspergillosis (CAPA) has been reported in about 30% of critically ill patients admitted to Intensive Care Units (ICUs) [4]. It usually appears 11 to 21 days after the onset of COVID-19 infection [5]. Some reports demonstrate a mortality rate near 50% after CAPA infection [6].

We hereby report a rare case of severe CAPA—and interesting related diagnostic images via lung ultrasound, CT scan, and bronchoscopy—presenting as IPA and IATB during hospitalization for COVID-19 in a critically ill patient who developed a further Aspergillus infection after home discharge.

## 2. Case

On day 0, a 67-year-old man was admitted to the Emergency Department (ED) after 3 days of persistent fever, confusion, and dyspnea. He tested positive for SARS-CoV-2 by polymerase chain reaction (PCR) nasal swab the week before ED admission. His medical history was consistent with arterial hypertension, type II diabetes mellitus poorly controlled, and hypothyroidism. He received his second mRNA COVID-19 vaccine six months before ED admission.

The patient was in severe respiratory distress, so non-invasive ventilation (NIV) through a full face mask was initiated. However, on day 2, orotracheal intubation and invasive mechanical ventilation were required due to the rapid worsening of respiratory gas exchanges and NIV intolerance. Despite optimized, protective mechanical ventilation, he remained hypoxemic (PaO_2_/FiO_2_ < 150 mmHg) and underwent, for this reason, two cycles of prone positioning on days 15 and 17.

Intravenous therapy with dexamethasone 6 mg daily was administered for 12 days. Gas exchange improved slowly, and on day 15, the patient was still mechanically ventilated in pressure support mode (pressure support 12 cmH_2_O, positive end-expiratory pressure 11 cmH_2_O) with a high inspired fraction of oxygen (FiO_2_ 70%). Bronchoalveolar lavage (BAL) was performed on day 17. It demonstrated branched septate hyphae suggestive of Aspergillus spp. at microscopic examination subsequently identified through PCR as Aspergillus flavus. The Galactomannan index on BAL was 3.28, while it was negative on serum. In addition, a culture examination of BAL tested positive for multidrug-resistant Pseudomonas aeruginosa (10^6^ UFC/mL). All results were interpreted according to Eucast European standards. Therapy with intravenous voriconazole (loading dose of 6 mg/kg for two doses followed by 4 mg/kg twice daily) and meropenem 1 g every 8 h was started through continuous infusion. Due to difficult and prolonged weaning from mechanical ventilation, a surgical tracheostomy was performed on day 20.

The antifungal treatment was continued for 60 days. Multiple antibiotic courses were performed during his hospital stay as well, according to microbiological tests (see Figure 1).

His nasal swab tested negative for SARS-CoV-2 on day 37. After two months of ICU stay, the patient’s respiratory condition improved, and he was successfully discharged to the intensive rehabilitation unit with a tracheostomy, spontaneously breathing, from where he was successfully sent home in acceptable clinical condition on day 73.

Eight weeks after hospital discharge (day 127), the patient returned to the ED with massive hemoptysis. He was in severe respiratory distress and hypoxemic (SpO_2_ 90% with 100% oxygen mask). His blood tests showed mild anemia (Hb 9 mg/dL) and an increased C-reactive protein. The patient’s conditions required invasive mechanical ventilation through a tracheostomy. A fiberoptic bronchoscopy was performed to clean airways from bright red blood clots. An obstructive and active bleeding lesion in the left main bronchus was identified, as shown in the video clip (Appendix A). Serum β-D-Glucan was 1.49 ng/mL (normal values < 0.08 ng/mL), and the Galactomannan index in BAL was positive. BAL tested positive for Aspergillus flavus. Microbiological findings from BAL are shown in Figure 1. A chest CT scan performed in the emergency department showed a complete collapse of the left upper lobe and lingula and partial collapse of the left lower lobe, with main bronchial branches obstructed by clots (arrow in Figure 2A). A pseudo-aneurysm was also found in the upper anterior left lobe (arrow in Figure 2B).

The patient was transferred to the ICU. Intravenous voriconazole was re-started at 400 mg twice a day in association with intravenous meropenem 2 g every 8 h. Multiple bronchoscopies during ICU stay were required to clean the airways and to allow for adequate ventilation. On day 131, during this second ICU stay, a bedside lung ultrasound was performed. It detected an oval hypoechoic mass of about 5.2 cm × 4.4 cm in the left upper anterior zone, characterized by some hyperechoic spots inside (Figure 3, yellow box), suggestive of hematoma. This finding was confirmed by a new following chest CT scan, which showed a hyperdense blood mass in the site of the previous pseudo-aneurysm (Figure 3). 

Partial re-expansion of the left upper lobe and lingula with the clearing of bloody material through bronchoscopy, thrombosis of segmental arterial branches of lingula and sub-segmental branches for the left upper lobe, and other lesions suggestive of aspergilloma were also depicted (Figure 4). 

Anticoagulant therapy with enoxaparin 4000 IU every 12 h was started. After 7 days, his clinical conditions markedly improved, and the patient was weaned off mechanical ventilation. On day 134, he was discharged from ICU.

## 3. Discussion

COVID-19 critically ill patients present mostly pulmonary and neurological involvement [7]. However, bacterial and fungal co-infections are very frequent. CAPA is a severe co-infection in COVID-19 patients admitted to ICU that worsens outcomes [8]. 

Many risk factors for aspergillosis have been identified in critically ill COVID-19 patients: severity of the disease per se as reflected by high APACHE II or SOFA score, severe lymphopenia, cytokine storm, need for invasive mechanical ventilation, use of steroids or immunosuppressive drugs such as tocilizumab and anakinra, chronic pulmonary diseases, and HIV [9]. During the first ICU stay, our patient was in severe condition, needing mechanical ventilation; he received a high dose of dexamethasone for 12 days and had uncontrolled diabetes before ICU admission. All these factors could have led to Aspergillus flavus primary pulmonary infection. Diagnosis of probable CAPA tracheobronchitis requires direct bronchoscopic observation of tracheobronchial lesions, as previously described, and mycological evidence such as positive BAL galactomannan, positive serum β-D-Glucan, or a positive culture for Aspergillus spp. In addition, there must be a pulmonary infiltrate or nodule, preferably documented by a chest CT scan, cavitating infiltrate (not attributed to another cause), or both, combined with mycological evidence [10]. In our case, the criteria for CAPA were fulfilled according to the classification defined by Koheler and colleagues, and voriconazole was given accordingly [10]. 

Once Aspergillus spp. infection is probable or proven, adequate antifungal treatment should be promptly started. It is of utmost importance because any delay in antibiotic administration increases mortality in ICU [11]. Moreover, it is also fundamental that the duration of antifungal treatment prolongs up to 6–12 weeks. 

However, as described by Kakamad et al., pulmonary aspergillosis is a serious complication of COVID-19 patients and may not respond well to adequate medical therapy requiring resection as the last and most effective strategy to control the disease [12].

In our case, the timing, dosage, and duration of voriconazole administration were adequate. However, the patient developed an IATB secondary to Aspergillus flavus reinfection. Little is known about IATB in critically ill patients. IATB appears as plaques, pseudomembranes, or ulcers in the trachea or the main bronchi. In this light, safety concerns about performing bronchoscopy, particularly during the early phase of the pandemic, may have led to underreporting of this type of Aspergillus spp. manifestation [11]. We argue that the Aspergillus flavus reinfection in our case could be due to different reasons: the patient experienced a very long and severe COVID-disease that required an ICU stay lasting more than two months. This may have favored the so-called prolonged inflammatory and catabolic syndrome with persistent reduced immune system function that, in turn, could have resulted in chronic pulmonary aspergillosis [13]. Therefore, we cannot exclude the formation of chronic cavitary pulmonary aspergillosis or latent aspergillosis infection that turns into tracheobronchial spreading and angio-invasion with bloodstream dissemination once the cavity has broken. Besides Aspergillus findings, pulmonary thromboembolism supports the angio-invasion event in this case [14]. 

Radiological imaging is important in the diagnosis of CAPA. However, during the peak of COVID-19 contagions, transporting critically ill patients was difficult due to hospital reorganization and was not recommended to reduce the viral spread [15].

Interestingly, in our case, the suspicion was raised by lung ultrasound.

Lung ultrasound is a non-invasive repeatable bedside imaging tool available in the ICU. During the COVID-19 pandemic, lung ultrasound has been used extensively [16]. Unfortunately, the aspergillus lesion is not specific at the ultrasound exam, and it can be detected only if it reaches the pleural surface. On the contrary, a central lesion cannot be seen with ultrasound. In our case, the lesion was superficial in the left upper lobe, leading us to further investigate the patient’s lesion with a CT scan.

Despite many advanced ventilatory monitoring tools, weaning from invasive ventilation may be difficult in CAPA patients, as demonstrated by our case [17]. Although our patient presented a life-threatening condition during the second hospitalization, his outcome was favorable. Noteworthy, the suspicion and search for fungal infections were considered since the earliest phase of the disease (the first ICU stay). Nevertheless, despite the prompt administration of the antifungal treatment, the clinical course deteriorated due to CAPA-related complications (the second ICU stay). Therefore, we strongly encourage a high degree of attention to be paid to fungal complications, even after COVID-19 viral recovery and ICU discharge. This is particularly relevant in patients with predisposing conditions, such as poorly controlled diabetes or prolonged ICU stay.

Considering that invasive aspergillosis could require prolonged follow-up, a possible strategy to reduce the risk of recurrence could be the repetition of serum biomarkers after prolonged adequate antifungal therapy. 

Finally, one important aspect that should be taken into consideration to reduce the risk of fungal infection is environmental monitoring. Evidence that regular surveillance and cleaning could reduce or eliminate the fungal contamination of indoor air has been described [18].

Despite the difficulty of monitoring indoors to limit viral spread during the COVID-19 pandemic, this issue should be pursued whenever possible [19]. 

## Figures and Tables

**Figure 1 healthcare-10-02483-f001:**
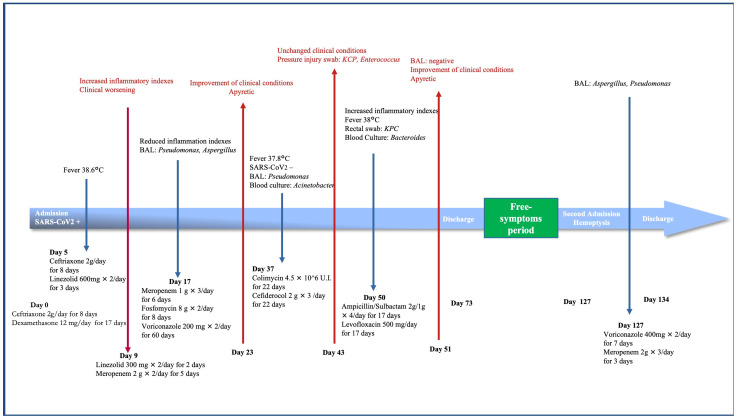
Timeline of disease progression and patient’s clinical course. Legend: thin red arrows pointing up show the response to the treatment initiated due to changes within the clinical condition of the patient, which are represented by thin blue arrows pointing down.

**Figure 2 healthcare-10-02483-f002:**
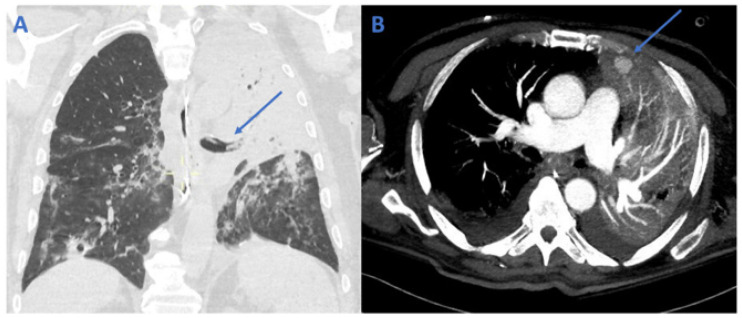
Chest CT scan at emergency department admission. Legend: complete collapse of left upper lobe and lingula and partial collapse of left lower lobe are shown, with main bronchial branches obstructed by clots (arrow in (**A**)). A pseudo-aneurysm was also found in the upper anterior left lobe (arrow in (**B**)).

**Figure 3 healthcare-10-02483-f003:**
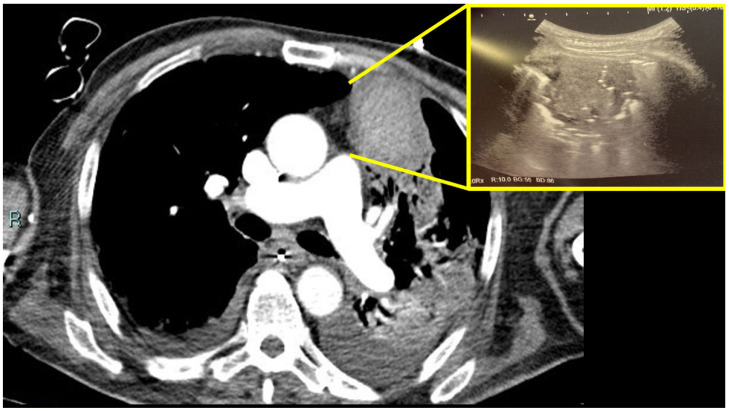
Lung bedside ultrasound evaluation and chest CT scan during ICU stay. Legend: with ultrasound, oval hypoechoic mass of about 5.2 cm × 4.4 cm in left upper anterior zone, characterized by some hyperechoic spots inside (Figure 3, yellow box), suggestive of hematoma. This finding was confirmed by chest CT scan, which showed a hyperdense bloody mass.

**Figure 4 healthcare-10-02483-f004:**
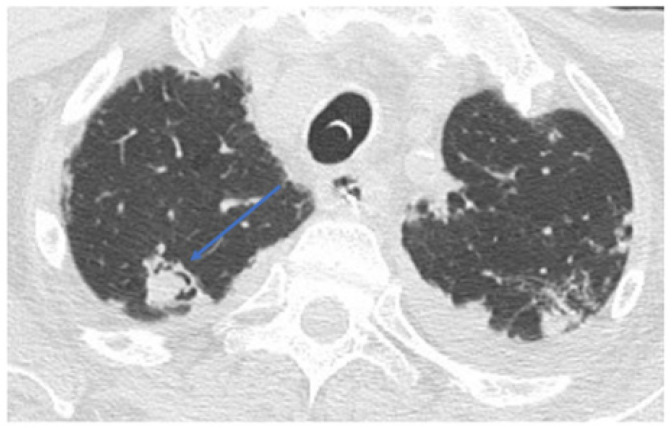
CT chest scan showing a lesion suspected as aspergilloma, as pointed out by blue arrow. Legend: blue arrow shows lesions suggestive of aspergilloma.

## Data Availability

Not applicable.

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
