# Peer review of "Severe Recurrent COVID-Associated Pulmonary Aspergillosis: A Challenging Case"

_healthcare, 2022, doi:10.3390/healthcare10122483_

Round 1

Reviewer 1 Report

The clinical case described by the authors stimulates interest, however we believe it is appropriate to underline some details.

The diagnosis of post-covid-19  invasive pulmonary aspergillosis has already been described (Int J Surg Case Rep 2021 May;82, SAGE Open Med Case Rep 2022 May 25). Therefore the authors should restructure the discussion by emphasizing which new stimuli can be highlighted by the description of their clinical case.

Was the antifungal therapy suspended at the first negative result of the fungal antigen? Was a new dosage performed in order to optimize the sensitivity of the results? It would be useful to clarify.

Which theraphy was prescrobed to the patient at the time of the first discharge? Has the patient continued potentially immunosuppressive therapy at home? It would be useful to clarify.

Author Response

Reviewer 1

The clinical case described by the authors stimulates interest, however we believe it is appropriate to underline some details.

The diagnosis of post-covid-19 invasive pulmonary aspergillosis has already been described (Int J Surg Case Rep 2021 May;82, SAGE Open Med Case Rep 2022 May 25). Therefore, the authors should restructure the discussion by emphasizing which new stimuli can be highlighted by the description of their clinical case.

Was the antifungal therapy suspended at the first negative result of the fungal antigen? Was a new dosage performed in order to optimize the sensitivity of the results? It would be useful to clarify.

Which therapy was prescribed to the patient at the time of the first discharge? Has the patient continued potentially immunosuppressive therapy at home? It would be useful to clarify.

Author’s reply: Thank you for your suggestions. We are aware that invasive pulmonary aspergillosis has already been described. However, we would like to point out mainly the aspergillosis tracheobronchial invasion as recurrent infection after adequate (as it regards timing and dosage according to current guidelines) antifungal treatment. This event is rare but demonstrates that invasive aspergillosis could require prolonged follow-up.

As correctly pointed-out by the reviewer, a possible strategy to reduce the risk of recurrence could be to repeat serum biomarkers also after prolonged antifungal therapy and we reported that in the discussion.

Second, we searched for the suggested references but we found only one and added the following: “Kakamad FH, Mahmood SO, Rahim HM, Abdulla BA, Abdullah HO, Othman S, Mohammed SH, Kakamad SH, Mustafa SM, Salih AM. Post covid-19 invasive pulmonary Aspergillosis: A case report. Int J Surg Case Rep. 2021 May; 82:105865. doi: 10.1016/j.ijscr.2021.105865. Epub 2021 Apr 6.”

Reviewer 2 Report

General comments

=============

Thank you for letting me peer-review your work! This paper is an interesting case of severe recurrent covid-associated pulmonary aspergillosis. I’m sorry I could not watch the video clip through the review system.

Specific comments

=============

Minor comments

---------------------

1. Please clarify the figure 1, especially in the vertical axis and the arrows.

2. If available, please add the imaging studies in initial admission and compare them between the first discharge and the second admission.

3. If available, please add the initial HbA1c.

4. Please add other risk factors of invasive aspergillosis, including chronic pulmonary diseases and HIV.

5. Please add to the discussion the lung ultrasound for invasive aspergillosis.

Author Response

Reviewer 2

General comments

=============

Thank you for letting me peer-review your work! This paper is an interesting case of severe recurrent covid-associated pulmonary aspergillosis. I’m sorry I could not watch the video clip through the review system.

Author’s reply: thank you for your request we reuploaded the file.

Specific comments

=============

Minor comments

---------------------

  1. Please clarify the figure 1, especially in the vertical axis and the arrows.

Author’s reply: For a better explanation, the figure represents the hospital time course of the patient. Red thin arrows pointing up show the response to the treatment initiated due to changes within the clinical condition of the patient which are represented by thin blue pointing down arrows.

And as for reviewer request, we added this in the figure legend.

  1. If available, please add the imaging studies in initial admission and compare them between the first discharge and the second admission.

Author’s reply: Thank you for your suggestion. The timing of the CT was different, but following your request we attach here a picture that represents the patient’s lung situation close to the admission.

  1. If available, please add the initial HbA1c.

Author’s reply: We apologize but HbA1c was not done.

  1. Please add other risk factors of invasive aspergillosis, including chronic pulmonary diseases and HIV.

Author’s reply: Thank you for this suggestion we include this in the discussion section as follow: “Many risk factors for aspergillosis have been identified in critically ill COVID-19 patients: severity of the disease per se as reflected by high APACHE II or SOFA score, severe lymphopenia, cytokine storm, need for invasive mechanical ventilation, use of steroids or immunosuppressive drugs such as tocilizumab and anakinra, chronic pulmonary diseases and HIV.”

  1. Please add to the discussion the lung ultrasound for invasive aspergillosis.

Author’s reply: According to your suggestion we added it. Lung ultrasound is a non-invasive repeatable bedside imaging tool available in ICU. During the COVID-19 pandemic lung ultrasound has been used extensively. Unfortunately, the aspergillus lesion is not specific at the ultrasound exam and can be detected only if they reach the lung surface. On the contrary central lesion cannot be seen with ultrasound.  In our case the lesion was superficial in the left upper lobe and led us to further investigate the patient’s lesion with CT scan.

Reviewer 3 Report

The authors present a case report on the severe recurrent covid-associated pulmonary aspergillosis.

Their study is based on a rare case of severe COVID-19 associated pulmonary aspergillosis (CAPA) and the diagnostic images with lung ultrasound, CT scan and bronchoscopy, presenting as invasive pulmonary aspergillosis (IPA) and invasive tracheobronchial aspergillosis (IATB) during hospitalization for COVID-19 in a critically ill patient who developed a further Aspergillus infection after home discharge.

The authors should check all the acronyms and specify, at least the first time in full, what they correspond to.

-Figure 1: The authors should improve the legend especially regarding the colors of the arrows and make it less chaotic. There are too much information on it. Furthermore, why do arrows go up or down? Please specify the reasons

-Figure 2-3-4: The authors should improve the legend and specify what arrows indicate

Author Response

Reviewer 3

The authors present a case report on the severe recurrent covid-associated pulmonary aspergillosis.

Their study is based on a rare case of severe COVID-19 associated pulmonary aspergillosis (CAPA) and the diagnostic images with lung ultrasound, CT scan and bronchoscopy, presenting as invasive pulmonary aspergillosis (IPA) and invasive tracheobronchial aspergillosis (IATB) during hospitalization for COVID-19 in a critically ill patient who developed a further Aspergillus infection after home discharge.

The authors should check all the acronyms and specify, at least the first time in full, what they correspond to.

-Figure 1: The authors should improve the legend especially regarding the colors of the arrows and make it less chaotic. There is too much information on it. Furthermore, why do arrows go up or down? Please specify the reasons

Author’s reply: We just try to give a more readability sense to the figure, however following the reviewer request we try to improve it.

-Figure 2-3-4: The authors should improve the legend and specify what arrows indicate

Author’s reply: Following the reviewer suggestion we tried to improve the legend of the figures as following:

Figure 2: Legend: Complete collapse of left upper lobe and lingula and partial collapse of left lower lobe are shown, with main bronchial branches obstructed by clots (arrow in figure 2 A). A pseudo-aneurysm was also found in the upper anterior left lobe (arrow in figure 2 B).

Figure 3: Legend: With ultrasound, ovalar hysoechoic mass of about 5.2 cm x 4.4 cm in left upper anterior zone, characterized by some hyperechoic spots inside (Figure 3, yellow box), suggestive for hematoma. This finding was confirmed by chest CT scan which showed an hyperdense bloody mass.

Figure 4: Legend: blue arrow shows lesions suggestive for aspergilloma.

Reviewer 4 Report

Dear colleagues,

The article provides very interesting and useful data. 

Author Response

Thank you!